# Introducing SPINS: A shared public visualization library of neuroanatomical structures

**Wali Sidiqyar**[*1]       WALI.SIDIQYAR@VANDERBILT.EDU
[1] *Brentwood High School, Nashville, TN, USA*

**Gaurav Rudravaram**[*2]       GAURAV.RUDRAVARAM@VANDERBILT.EDU
[2] *Vanderbilt University, Nashville, TN, USA*

**Elyssa M. McMaster**[2]       ELYSSA.M.MCMASTER@VANDERBILT.EDU
**Trent M. Schwartz**[2]       TRENT.SCHWARTZ.1@VANDERBILT.EDU
**Adam M. Saunders**[2]       ADAM.M.SAUNDERS@VANDERBILT.EDU
**Kurt G. Schilling**[3]       KURT.G.SCHILLING.1@VUMC.ORG
[3] *Vanderbilt University Medical Center, Nashville, TN, USA*
**Bennett A. Landman**[2,3]       BENNETT.LANDMAN@VANDERBILT.EDU

## Abstract

Brain atlases help researchers and clinicians understand brain structure, interpret results, and communicate findings. However, no publicly accessible, precomputed resource currently exists for comparing regions across brain atlases in 3D. To address this gap, we introduce a resource that systematically visualizes regions across atlases, lowering barriers for researchers and clinicians to explore and communicate brain anatomy. We developed an automated pipeline that renders brain regions across multiple atlases, including gray matter, white matter, and functionally relevant regions. Our website offers page-based navigation of 210 regions, each featuring an AI-generated overview, multiple glass brain videos, and tri-planar visualizations. The complete visualization library is open source and freely accessible at https://masilab.github.io/SPINS/.

**Keywords:** Neuroimaging visualization, brain atlas, open-source

## 1. Introduction

Magnetic resonance imaging (MRI) allows for the acquisition of high-resolution data to observe brain structure and function (Jones, 2010). Brain atlases, or sets of spatial anatomical maps based on expert and histological knowledge of brain regions, provide insight into the relationships among brain regions. Analysis depends on the atlas specified for every task, as the cortical structure, white matter connectivity, histological architecture, and functional organization require different label sets of interest (Carter, 2019). Yet, full comprehension of the relationships between brain regions cannot always be inferred from two-dimensional static slices, while three-dimensional volume renderings for all anatomical labels of interest are technically challenging and computationally expensive to produce.

The neuroimaging community has developed a number of brain image rendering tools, such as FSL (Jenkinson et al., 2012), MRtrix3 (Tournier et al., 2012), and OpenDIVE (Saunders et al., 2025), however, none offer a publicly accessible, precomputed resource to

---

[*] Contributed equally

visualize atlas regions in three dimensions. Building on this need, we introduce SPINS, a Shared Public vIsualization library of Neuroanatomical Structures. SPINS is a curated collection of prerendered, open-source visualizations spanning multiple atlas types and visualization styles with the goal of expanding these visualizations to all defined neuroanatmical atlases in the future.

## 2. Methods

In this work, we demonstrate the utility of SPINS across three atlases: BrainCOLOR (Klein and Tourville, 2012; Huo et al., 2019), a cortical gray matter parcellation; Pandora-TractSeg (Hansen et al., 2021; Wasserthal et al., 2018), a white matter tract atlas; and the Yeo 17-network atlas (Yeo et al., 2011), a functional parcellation. For each atlas, we used a T1-weighted image and a corresponding segmentation file, both registered to the same space. To extract each region, we created a binary mask that isolates the region from the segmentation. We then upsampled these binary masks to 0.5 mm isotropic voxel resolution and applied Gaussian smoothing to minimize voxel-grid artifacts. Using the marching cubes algorithm, we converted each smoothed mask into a surface mesh. Next, we brought each mesh into world coordinates by applying the segmentation's affine matrix. We rendered each region on a transparent brain outline generated by upsampling, smoothing, and dilating the whole-brain segmentation, then subtracted the pre-dilation volume to highlight only the brain's outer shell. For atlases with overlapping regions, each label appears on this common outline. In addition to rendering all regions together for each atlas, we also rendered and recorded each region separately on transparent outlines, producing both black and white background versions. Each region's mesh is colored according to the atlas's recommended color scheme. Finally, we exported videos as MP4 and GIF for easy embedding in websites or presentations (Figure 1).

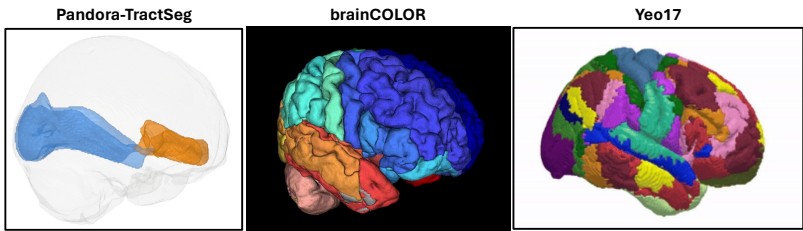

Figure 1: The spinning brain video renderings on the main atlas page display all regions simultaneously, providing an overview of how parcellations are distributed and how densely each atlas labels the brain. For Yeo17 and brainCOLOR, which contain no overlapping regions, all regions are rendered at once. For Pandora, which contains overlapping regions, each region is rendered individually with the subsequent region gradually fading in.

Beyond the atlas-level video, we rendered each region separately for every atlas against a transparent brain outline. To generate each region's surface mesh, we up-sampled the binary mask to 0.5 mm isotropic voxel resolution, Gaussian-smoothed to reduce voxel grid artifacts, and applied marching cubes to extract the surface. The transparent brain background is generated separately using the same up-sampling and smoothing pipeline, with heavier smoothing and two dilation passes; the pre-dilation volume is subtracted from the dilated result to isolate only the outer portions of the brain. This generates an image with visible cortical folding, but the inner surfaces are suppressed, making the shell appear as an outline

when rendered at a low transparency setting. We provide two versions of this rendering (a black background and a white background). In each case, the region is rendered in the color recommended by the lookup table provided by the atlas's creators. (Figure 2 C). The full video of the spinning brain is exported as both an MP4 and a GIF for easy embedding directly in websites or presentations without requiring a video player.

We also generated two images for each region. For the first image, we overlaid the region's binary mask onto sagittal, coronal, and axial slices of the T1-weighted image, passing each slice through the region's centroid to provide anatomical context. The second image is a glass-brain projection, showing the region's three orthogonal orientations superimposed on a transparent outline of the brain. This outline is created by up-sampling, smoothing, and dilating a skull-stripped, binarized T1-weighted image; we then subtract the pre-dilation volume from the dilated image to extract the outer brain shell. Both images are exported at 300 DPI for immediate use in figures or publications. Finally, each region is accompanied by an AI-generated atlas and region description, with links to relevant Wikipedia pages that provide additional context for users unfamiliar with the anatomy. (Figure 2 B). All visualizations, images, and region descriptions are publicly released as part of SPINS v1.0 and are freely accessible.

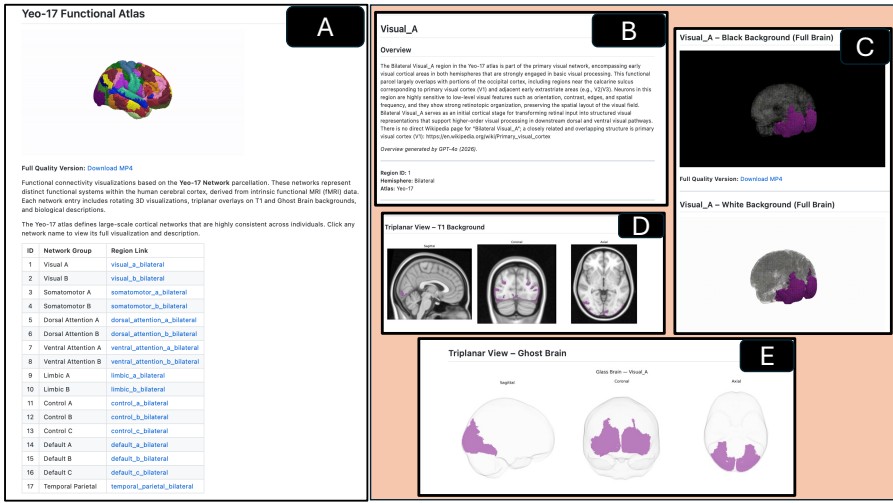

Figure 2: Overview of the SPINS website demonstrated using the Yeo-17 functional atlas and the Visual_A region. (A) Atlas homepage displaying a spinning brain video of all regions, a GPT-4o generated atlas description, and a table of regions with links to individual region pages. (B) GPT-4o generated natural language overview of the region's anatomical and functional context. (C) 3D spinning brain visualization with the region of interest highlighted in its atlas lookup table color (D) Triplanar segmentation overlay on the T1 image (E) Glass brain triplanar projection from the binary mask

## Acknowledgments

We would like to acknowledge support from NIH 5U01DA055347-03, P50HD103537, SFP302547 and 1R01EB017230-01A1. This work was supported by the Alzheimer's Disease Sequencing Project Phenotype Harmonization Consortium (ADSP-PHC) that is funded by NIA (U24 AG074855, U01 AG068057 and R01 AG059716).

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
