# OpenReview forum: "Introducing SPINS: A shared public visualization library of neuroanatomical structures"
_MIDL.io/2026/Short_Papers — MIDL 2026 - Short Papers Poster_

### Official Review · Reviewer_fbts · 2026-04-24
**Good Paper**

**Rating:** 4
**Confidence:** 4

**Review:**

Overall, I like the work. It is a perfect example of a short-paper contribution at MIDL that does not necessarily require a novel research idea but also supports open-source toolkits.

**Summary:**

The paper introduces a new visualization tool for a neuroimage atlas in three dimensions. It is an open-source toolkit that supports multiple atlas types and visualization styles: a cortical gray matter parcellation, a white matter tract atlas, and a functional parcellation. The open-source nature of the project, good visualization and corresponding AI-generated overview of different regions make it a useful tool for teaching.

**Strengths:**

This is a simple yet effective toolkit that would be useful for various researchers in the neuroimage analysis field. Open-source nature, publicly available atlases, and ease of visualization make this toolkit an attractive tool for the community.

**Weaknesses:**

* No major weaknesses; however, it would be a good idea to see if the visualization capabilities could be provided for custom atlases. It will make the tool much more useful in the future for various research purposes.
* Rather than a static GIF or MP4 file, it would be useful to make these atlases more interactive, where users could select a region based on what they are seeing rather than the name of the structure.

**Justification Of Rating:**

Paper is a solid work as a toolkit for visualization. However, there are a couple of points that might be good to address. I have listed them in the weakness section. Considering everything, I am recommending weak accept.

---

### Decision · Program_Chairs · 2026-05-08

Accept (Poster)